# An Internal Marking Method for Adult *Spodoptera frugiperda* Smith Using an Artificial Diet Containing Calco Oil Red N-1700

**DOI:** 10.3390/insects15080561

**Published:** 2024-07-25

**Authors:** Shishuai Ge, Bo Chu, Xiaoting Sun, Jiajie Ma, Xianming Yang, Kongming Wu

**Affiliations:** 1School of Resources and Environment, Henan Institute of Science and Technology, Xinxiang 453003, China; geshishuai68@163.com; 2State Key Laboratory for Biology of Plant Diseases and Insect Pests, Institute of Plant Protection, Chinese Academy of Agricultural Sciences, Beijing 100193, China; chubo907@163.com (B.C.); sxiaoting1993@163.com (X.S.); m15736725579@163.com (J.M.); zqbxming@163.com (X.Y.); 3College of Plant Protection, Henan Agricultural University, Zhengzhou 450002, China; 4College of Tropical Crops, Hainan University, Haikou 570228, China; 5State Key Laboratory of Cotton Bio-Breeding and Integrated Utilization, School of Life Sciences, College of Agriculture, Henan University, Kaifeng 475004, China

**Keywords:** fall armyworm, two-sex life table, flight performance, dye marking, mark–release–recapture

## Abstract

**Simple Summary:**

*Spodoptera frugiperda* Smith (fall armyworm; FAW) (Lepidoptera: Noctuidae) has invaded and spread across Africa, Asia, and Oceania for several years. Conducting mark–release–recapture (MRR) experiments will help clarify the migration behavior and patterns of the FAW in newly invaded areas. Therefore, we researched a suitable internal dye marking method for the FAW by incorporating different concentrations of Calco Oil Red N-1700 into an artificial diet. The biological indicators (developmental duration, reproductive parameters, flight ability, etc.) of the FAWs fed a diet containing a 0.2% concentration of Calco Oil Red N-1700 were normal, and both female and male adults exhibited clear marking colors. This study can promote the development and application of MRR experiments for the FAW, and will help formulate effective pest management strategies.

**Abstract:**

As a migratory invasive pest, *Spodoptera frugiperda* (fall armyworm, FAW) has recently posed a serious threat to food security in newly invaded areas (especially in Africa and Asia). Understanding its migration (or dispersal) patterns in newly invaded areas is crucial for regional forecasting and management efforts. By screening an appropriate marking technique to conduct mark–release–recapture (MRR) experiments, the migration patterns of the FAW can be effectively studied. In this study, we added different concentrations of Calco Oil Red N-1700 (an oil-soluble marker) to a self-made artificial diet and assessed the rearing and marking efficacy. The results indicated that a concentration of 0.2% of Calco Oil Red N-1700 in the diet was optimal for marking adult FAWs. The biological indicators (e.g., developmental duration, reproductive parameters, and flight ability) of FAWs fed this diet were basically consistent with those of FAWs fed a normal diet, with a larval stage of 15.46 days, a pupal stage of 9.81 days, a pupal mass of 278.18 mg, an adult longevity of 15.41 days, and an egg deposition count of 1503.51. Meanwhile, the flight distance, duration, and velocity were 24.91 km, 7.16 h, and 3.40 km/h, respectively (12 h tethered-flight tests), without difference with the control. Females and males exhibited distinctive marking colors (red or pink) that persisted for at least 5 and 9 days, respectively. This study developed an economically effective internal marking method for the adult FAW, laying the foundation for conducting MRR experiments. This will help clarify the migration behavior and routes of the FAW, providing a scientific basis for formulating effective pest management strategies.

## 1. Introduction

The fall armyworm (FAW; *Spodoptera frugiperda*) originated in the Americas [1] and has invaded and spread to Africa, Asia, and Oceania [2,3,4], posing a serious threat to food production and security in newly invaded areas. The pest is also reported to be present in the EU, with distribution restricted to Cyprus (European Food Safety Authority pest survey card, https://storymaps.arcgis.com/stories/06fb4d48431a409eadfa2413544d275e, (accessed on 18 July 2024)). The invasive FAW population has been identified as a maize strain [5], and the larvae feed on a variety of important food and economic crops, such as maize, wheat, and sugarcane [1,6]. The FAW damaged 1.125 and 1.278 million hectares of crops in China in 2019 and 2020, respectively [7], with an annual loss of USD 940 million in Africa alone [8]. Even if the FAW has colonized for several years, chemical pesticides are still mainly used to control the FAW in newly invaded areas, resulting in a series of problems, such as significant increases in control costs, environmental pollution, and impacts on food safety [9,10].

The FAW is a typical migratory insect [11,12], and migration (or dispersal) is a behavioral factor that leads to regional infestations and host shifts. FAW adults have the ability to migrate across borders or regions, such as the FAW populations from the Indochina Peninsula, northeastern India, and Nepal, which can migrate into China with the monsoon [13,14]. It has been estimated that FAW adults can enter the southern Sahara from western parts of Africa in one single night or over a few consecutive nights [15]. Zhou et al. [16] captured FAW adults by using searchlight traps set up on Yongxing Island in the South China Sea and found that FAWs from the Philippines can cross the South China Sea to mainland China through trajectory analysis. Wu et al. [17] speculated that FAW adults captured in Japan may originate from southern China. The ability of the FAW to migrate across regions increases the complexity associated with its forecasting and regional management [13,14]. The FAW is listed as a quarantine and priority pest in the EU, and many efforts are being made to prevent its introduction into and spread within the EU, e.g., performing annual surveys and standardizing the surveillance methods to monitor the emergency situation (EFSA pest survey card). In addition, the United Nations FAW Monitoring and Early Warning System (FAMEWS) online tool has been developed to monitor the spread of FAW adults. Our team is also engaged in research to better prevent and control this migratory pest in China.

Understanding FAW migration patterns in newly invaded areas is the basis for accurately predicting population dynamics and guiding control efforts. Mark–release–recapture (MRR) experiments are widely used in insect migration studies and help to clarify the migration routes and patterns of migratory insects [18]. For example, Showers et al. [19] used the same method to confirm that *Agrotis ipsilon* migrates with the southward near-surface airflow in late summer and autumn, providing some insight into the migration patterns and routes of the insect in North America. The selection of an appropriate marking method is crucial for MRR experiments. There are many methods for marking insects, such as dust, dye, chemical, and genetic marking, which should be selected based on actual needs and insect characteristics. The ideal marker should not interfere with the normal life activities of insects, such as their growth, development, feeding, mating, and flight, and should be stable, long-lasting, environmentally benign, inexpensive, and easy to use [20].

Since the 1960s, significant advances have been made in the use of dye markers to mark insects, particularly with the development of the Sterile Insect Technique (SIT) and Area-Wide Integrated Pest Management (AW-IPM) [21,22]. Typically, the dye is mixed with alcohol and water at a certain ratio to form a marking solution that is used to mark the insect’s body surface by smearing or spraying [23]. However, the stability of this in vitro dye marking method is somewhat poor, and it is easily affected by environmental conditions in the field, causing the markers to be lost. The high alcohol content in the solution may also cause some damage to the test individuals. An in vivo dye marking method can effectively avoid these problems. Several oil-soluble markers (e.g., Calco Oil Red N-1700, Sudan Black B, Oil Soluble Blue, and Rhodamine B) can be added to insect diets for larvae to feed on, and the markers will accumulate in their body fluids or tissues [24,25]. Among these, Calco Oil Red N-1700 is a widely used marker that can produce highly visible and durable colors when added to the diets of some lepidopteran insects (e.g., *Pectinophora gossypiella* and *A. ipsilon*) [26,27].

In order to identify an appropriate marking method for the FAW and provide a comprehensive and intuitive study of its regional migration behavior and patterns, in this study, we investigated the effects of artificial diets containing different concentrations of Calco Oil Red N-1700 on the growth, development, survival, reproduction, and flight performance of the FAW. Then, we evaluated the marking effects at each concentration and determined the appropriate concentration of Calco Oil Red N-1700.

## 2. Materials and Methods

### 2.1. Study Insects

The FAWs were reared in the laboratory across 20 generations. The larvae were reared in transparent plastic containers (22 × 15 × 8 cm; 30–40 individuals per container) and fed an artificial diet based on soybean flour and wheat bran [28]. The pupae were separated and placed in glass culture dishes with moistened cotton. Following pupal emergence, male and female moths were placed in plastic buckets (diameter of 25 cm, height of 30 cm, and 15 L capacity) for mating. The moths were fed with a 10% *v*/*v* honey/water solution, and the buckets were covered with sterile gauze to collect egg masses. The FAWs were reared under controlled conditions at 25 ± 1 °C, 70% ± 10% RH, and a 16:8 h L:D photoperiod.

### 2.2. Preparation Method of Diets Containing Calco Oil Red N-1700

A self-made artificial diet for the FAWs was used, with the same diet formulation and preparation method as used by Ge et al. [29]. In the later stage of diet preparation (i.e., after the addition of other materials), 0.18 g (0.01% of the total diet weight), 0.9 g (0.05%), 1.8 g (0.1%), 3.6 g (0.2%), and 9 g (0.5%) of Calco Oil Red N-1700 (CAS: 4477-79-6, purity: 99%; Hubei Shishun Biotechnology Co., Ltd., Hubei, China) were added and stirred well to prepare artificial diets containing different concentrations of Calco Oil Red N-1700, with a normal diet (without Calco Oil Red N-1700) as the control (Appendix A).

### 2.3. Experimental Assays

#### 2.3.1. Effects of Different Diets on the Development, Survival, Reproduction, and Flight Performance of *Spodoptera frugiperda*

We collected egg masses deposited by adults from the laboratory colony and observed them daily until hatching. The diets containing varying concentrations of Calco Oil Red N-1700 were cut into small square pieces and placed in plastic cups (38 × 30 × 30 mm, 25 mL). Subsequently, we placed the newly hatched larvae in the cups and numbered them sequentially. To prevent mechanical damage to or death of the larvae caused by human operation, we added 2–3 larvae to each cup, and we left one randomly selected healthy larva in the cup on the following day. We transferred the other larvae to a new cup (1 larva/cup), which we fed in the same way until emergence and testing of flight performance. We renewed the diets every 2–3 days to avoid the influence of food freshness on the experimental results. We recorded the development and survival of the larvae daily until pupation. On the third day after pupation, we weighed the pupae on an electronic balance (0.0001 g, Mettler Toledo ME204; Beijing Haitian Youcheng Technology Co., Ltd., Beijing, China) since they were fragile in the early stage. We checked the pupae for survival daily until death or eclosion. Following adult eclosion, we recorded the sex of the adults and checked them for deformities. We recorded the longevity of any deformed adult as 1 day. Then, we paired healthy adults in disposable plastic cups (500 mL) (1 mating pair/cup), covered each plastic cup with sterile gauze for collecting egg masses, and placed a cotton ball immersed in a 10% *v*/*v* honey/water solution at the bottom of the cup to provide water and nutrients to the FAWs. On a daily basis, we refreshed the cotton ball and the gauze with egg masses. We replaced the cup if egg masses were deposited on the cup wall or bottom. We documented the preoviposition period, daily fecundity, oviposition period, egg hatchability, and adult longevity. We determined mating status (mated/unmated) via the presence of spermatophores in the female after death. For the experiments, we reared the FAWs in climate chambers (MGC-450HP; Yiheng Technology Instrument Co., Ltd., Shanghai, China) with 5000 Lx illumination at 25 °C, 75% RH, and a 16:8 h L:D photoperiod. Each diet treatment consisted of 80 FAW individuals with three replicates, giving a total of 240 individuals.

We transferred the remaining adults in each diet treatment to different cages and fed them a 10% *v*/*v* honey/water solution. Next, we individually attached 2-day-old healthy adults to flight mills (FXMD-24-USB flight mill; Jiaduo Science, Industry and Trade Co., Ltd., Henan, China) in artificial climate chambers according to the method described by Ge et al. [30]. The tests were conducted for 12 h, from 8 p.m. to 8 a.m. the following day (Chinese standard time). The chambers were kept completely dark and maintained at 25 °C and 75% RH. There were at least 30 test individuals for each treatment.

#### 2.3.2. Marking Location and Color Intensity

We also reared some FAWs fed diets containing varying concentrations of Calco Oil Red N-1700. We selected early emerged female and male adults at each concentration, extracted the fat bodies from the abdominal epidermis, and then measured their color intensity using a colorimeter (3nh, NR60CP; Shenzhen 3nh Technology Co. LTD., Shenzhen, China). The color intensity was represented by the Euclidean distance from white (DW value) using the methods of Prohens et al. [31] and Cericola et al. [32], and was calculated using the formula: DW = ((100 − L*)2 + a*2 + b*2)0.5. The values of L*, a*, and b* were obtained using a colorimeter, with the light source selected as D65, the measurement aperture as 4 mm, and the color space as CIEL*a*b*, which was measured in a dark environment. Since the fat bodies of a single adult did not meet the measurement conditions, we put the fat bodies of 2–3 individuals together as a group of samples for measurement. We repeated the measurements three times for each group, with five groups for both females and males.

#### 2.3.3. Duration of Marking Under the Optimal Concentration

Based on the above experimental results, the optimal concentration of Calco Oil Red N-1700 added to the diet was determined. Then, we reared a batch of FAWs at this concentration. The adults were placed in cages (30 × 30 × 30 cm) for mating and fed daily with a 10% *v*/*v* honey/water solution. We observed the coloration of tissues in female and male moths of different ages (1, 3, 5, 7, and 9 days old) under a microscope (TS-75X; Shanghai Shangguang New Optical Technology Co., Ltd., Shanghai, China). We counted the coloring rate at different ages, which was the proportion of individuals with marked colors to all test individuals. Each repetition consisted of 20–25 moths of each age group, repeated three times.

### 2.4. Data Analyses

A one-way ANOVA was used to analyze the effects of diets containing different concentrations of Calco Oil Red N-1700 on the growth, development, reproductive parameters, adult flight performance, color intensity of the fat bodies, and coloring rate of the FAWs. This was followed by Tukey’s HSD post hoc test if there were significant differences between the groups. Before the ANOVA, the data were first tested for normality using the Shapiro–Wilk test and for homogeneity of variance using Levene’s test. If the data did not meet the conditions, we performed an arcsine transformation; proportional data were arcsine square root transformed. We analyzed the interaction of diet and sex (with diet and sex as the fixed effects) on the development, pupal weight, and color intensity of the FAWs using a general linear mixed model. We used log-rank (survival analysis) testing to determine the differences and linear trends in the survival curves of the FAWs fed diets containing different concentrations of Calco Oil Red N-1700. The above statistical analyses were performed using SPSS software (version 26.0; SPSS software Inc., Chicago, IL, USA). We used TWOSEX-MSChart (version 2019.02.23) to examine the life tables (*s_xj_*, *l_x_*, *f_x_*, *m_x_*, *l_x_m_x_*, *e_xj_*) and demographic parameters (*R*_0_, *T*, *r*, *λ*) and a paired bootstrap test with 100,000 replications in TWOSEX-MSChart to accurately estimate the mean and standard error among the demographic parameters of the FAWs fed different diets [33,34,35,36]. Nonlinear regression analyses (Sigmoidal) of the color intensity (i.e., DW values) of the adult fat bodies at each concentration were performed using GraphPad Prism 8.0.2 (GraphPad Software Inc., San Diego, CA, USA) to obtain fitted curves and equations. The level of significance level was *p* < 0.05.

## 3. Results

### 3.1. Effects of the Diets on the Development, and Reproduction of Spodoptera frugiperda

#### 3.1.1. Development and Reproduction

Diets containing different concentrations of Calco Oil Red N-1700 affected the developmental period of the FAWs in each stage (except the egg stage) (Table 1). Specifically, except for 1st instar larvae (*F*_5, 1434_ = 1.474, *p* = 0.195), the developmental duration of each instar larvae (2nd instar larvae: *F*_5, 1195_ = 27.799, *p* < 0.001; 3rd instar larvae: *F*_5, 1434_ = 14.300, *p* < 0.001; 4th instar larvae: *F*_5, 1433_ = 7.904, *p* < 0.001; 5th instar larvae: *F*_5, 1433_ = 8.341, *p* < 0.001; 6th instar larvae: *F*_5, 1427_ = 21.193, *p* < 0.001), larval stage (*F*_5, 1427_ = 34.818, *p* < 0.001), pupal stage (*F*_5, 1357_ = 10.192, *p* < 0.001), egg-pupa stage (*F*_5, 1357_ = 22.913, *p* < 0.001), adult longevity (*F*_5, 1357_ = 4.411, *p* = 0.001), and total generations (*F*_5, 1357_ = 6.411, *p* < 0.001) significantly differed among the different treatments. The larval stage shortened with increasing concentrations of Calco Oil Red N-1700, with the shortest larval stage of 15.02 days in the 0.5% treatment, followed by the 0.2% and 0.1% treatments. The pupal stage of the FAWs in the 0.01% treatment was the longest (10.21 days), which was significantly longer than that of the control group, and there were no significant differences among the other treatments. The adult longevity and total generation of the FAWs in the 0.5% treatment were shorter than that of the control group, and there were no significant differences between the other treatments and the control group. There was a significant difference in the pupal mass (*F*_5, 1357_ = 5.205, *p* < 0.001) among the different treatments, with the pupal mass in the 0.5% treatment (276.77 mg) being significantly lower than that of the control group.

The diets greatly affected the reproduction of the FAWs (Table 1). The oviposition period (*F*_5, 580_ = 8.971, *p* < 0.001) among the different treatments differed significantly, and the oviposition period in the 0.1%, 0.2%, and 0.5% treatments was significantly shorter than that of the control group. There were no significant differences in the pre-oviposition period (*F*_5, 580_ = 1.792, *p* = 0.113), eggs deposited per female (*F*_5, 580_ = 1.612, *p* = 0.155), or mating rate (*F*_5, 12_ = 1.870, *p* = 0.174) among the different treatments.

The larval stage (*F*_1_ = 4.307, *p =* 0.038), pupal stage (*F*_1_ = 1557.424, *p* < 0.001), adult stage (*F*_1_ = 293.674, *p* < 0.001), and total generation (*F*_1_ = 509.295, *p* < 0.001) differed between the sexes among the different treatments, with no significant differences between the sexes in pupal mass (*F*_1_ = 0.100, *p* = 0.751). Overall, the larval stage, pupal stage, adult stage, total generation, and pupal mass of the females were smaller than those of the males (Figure 1). There were no significant diet × sex interactions in the larval stage (*F*_5_ = 1.564, *p* = 0.167), pupal stage (*F*_5_ = 1.408, *p* = 0.219), adult longevity (*F*_5_ = 1.007, *p* = 0.412), total generation (*F*_5_ = 0.991, *p* = 0.422), or pupal mass (*F*_5_ = 1.937, *p* = 0.085).

#### 3.1.2. Survival Rate, Fecundity, and Life Expectancy

Diets containing different concentrations of Calco Oil Red N-1700 greatly affected the age–stage survival rate (*s_xj_*) of the FAWs (Figure 2). There were significant differences in the larval survival rate among the different treatments (*F*_5, 12_ = 3.767, *p* = 0.028), and the larval survival rate in the 0.05%, 0.2%, and 0.5% treatments was significantly higher than that of the control group. There were no significant differences in the pupal survival rate (*F*_5, 12_ = 2.623, *p* = 0.079) or adult deformity rate among the different treatments (*F*_5, 12_ = 0.822, *p* = 0.557) (Table 2).

Survival analysis indicated that there were significant differences in the age-specific survival rate (*l_x_*) of the FAW population fed diets containing different concentrations of Calco Oil Red N-1700 (*χ*^2^ = 21.021, *df* = 5, *p* = 0.001), with a significant linear trend (*χ*^2^ = 13.971, *df* = 1, *p* < 0.001) (Figure 3). The maximum value of the age-specific fecundity of the female adults (*f_x_*) in the CK, 0.01, 0.05, 0.1, 0.2, and 0.5 treatments was 270.72, 270.39, 243.30, 260.30, 247.69, and 215.34, respectively. The age-specific fecundity (*m_x_*) and age-specific maternity (*l_x_m_x_*) curves indicated that the peak oviposition day in the 0.2% treatment appeared earliest. The maximum value of *m_x_* in the six groups occurred on days 33, 35, 33, 33, 32, and 34, respectively, and on days 33, 35, 33, 33, 32, and 34 for *l_x_m_x_*. The age–stage specific life expectancy (*e_xj_*) curve showed that the *e_xj_* in the 0.05% treatment was longer, with the shortest *e_xj_* for the 0.5% treatment (Figure 4).

#### 3.1.3. Life Table Parameters

Diets containing different concentrations of Calco Oil Red N-1700 greatly affected the life table parameters of the FAWs (Table 3). There was no significant difference in the net reproductive rate (*R*_0_) for the FAW population among the different treatments. The mean generation time (*T*) of the FAW population in the 0.01% treatment was significantly longer than that of the other treatments (0.01% and CK: *p* < 0.001; 0.01% and 0.05%: *p* = 0.002; 0.01% and 0.1%: *p* < 0.001; 0.01% and 0.2%: *p* < 0.001; 0.01% and 0.5%: *p* < 0.001). The FAW population in the 0.2% and 0.5% treatments had the highest intrinsic rate of natural increase (*r*) and finite rate of increase (*λ*), indicating that these two treatments are suitable for rapid population growth for FAW.

### 3.2. Effects of the Diets on the Flight Performance of Spodoptera frugiperda

The 12 h tethered-flight tests showed that the diets containing different concentrations of Calco Oil Red N-1700 had a significant effect on the flight performance of the FAWs (Table 4). The flight distance (*F*_5, 207_ = 4.038, *p* = 0.002) and duration (*F*_5, 207_ = 5.983, *p* < 0.001) differed among the different diet treatments, with no difference recorded for flight velocity (*F*_5, 207_ = 1.535, *p* = 0.180). The flight distance and duration of the moths in the 0.2% treatment were 24.91 km and 7.16 h, respectively, while they were 12.28 km and 3.35 h in the 0.5% treatment, respectively.

### 3.3. Coloring Location and Color Intensity of Females and Males Fed Different Diets

Observations showed no obvious color change in the reproductive system or intestines of the marked females and males compared to the control adults. The coloring location of Calco Oil Red N-1700 was mainly in the fat bodies, which appeared red or pink (Figure 5).

There were significant differences in the color intensity of the fat bodies of one-day-old adults fed diets containing different concentrations of Calco Oil Red N-1700 (females: *F*_5, 24_= 104.617, *p* < 0.001; males: *F*_5, 24_ = 53.507, *p* < 0.001). The color intensity (i.e., DW values) of both the males and females were significantly greater in the 0.2% and 0.5% treatments than in the other treatments, with no significant differences between the two treatments. Nonlinear regression was performed on the color intensity in each treatment, and the fitted curves were obtained, as shown in Figure 6. The color intensity (*F*_1_ = 28.685, *p* < 0.001) differed between the sexes among the different treatments, and there were no significant diet × sex interactions in color intensity (*F*_5_ = 0.530, *p* = 0.752).

### 3.4. Coloring Rate of Adults Fed Diets Containing 0.2% of Calco Oil Red N-1700 at Different Ages

Combined with the above results, we concluded that 0.2% of Calco Oil Red N-1700 is suitable for marking FAW adults without adversely affecting their growth, development, and flight performance, and the marking effect is excellent.

There was a significant difference in the coloring rate of the females fed diets containing 0.2% of Calco Oil Red N-1700 at different ages (*F*_4, 10_ = 72.948, *p* < 0.001) (Table 5). The coloring rate of one to five-day-old females was 100%, then gradually decreased with increasing age, reaching 65.56% at nine days old. Additionally, all males had obvious marking colors at one to nine days old.

## 4. Discussion

In a previous study, we developed a cost-effective artificial diet (based on wheat bran) for FAW larvae [29], significantly reducing the cost of rearing FAWs on a large scale. Based on the study, we added different concentrations of Calco Oil Red N-1700 to this artificial diet and assessed the rearing effects using a two-sex life table and tethered-flight tests, as well as the marking efficacy of each concentration. Then, we screened out a suitable internal marking method for FAWs. We found that a 0.2% concentration of Calco Oil Red N-1700 in the artificial diet was suitable for marking FAW adults; the marking effect was good and did not negatively affect the FAWs.

Calco Oil Red N-1700 has been successfully used as an internal marking dye for some lepidopteran insects (e.g., *P. gossypiella* and *A. ipsilon*), producing highly visible and long-lasting marking colors at appropriate concentrations without negatively affecting insect growth and development [26,27]. This study also indicated that an appropriate concentration of Calco Oil Red N-1700 does not inhibit the normal life activities of insects. The added concentration of Calco Oil Red N-1700 required by different insects may vary. For example, Li [37] found that more than 99% of *Helicoverpa assulta* females and males could be successfully marked at a concentration of 0.03%, while this study found that the appropriate addition concentration of Calco Oil Red N-1700 for FAWs was 0.2%. A lower concentration of Calco Oil Red N-1700 would affect the marking effect, and the marking color obtained by the insects would be lighter or not colored. Meanwhile, a higher concentration would affect the normal life activities of the insects. For example, the field experience of Hagler et al. [38] showed that marking insects with Calco Oil Red N-1700 could result in a small percentage of adults not retaining their marking color, and we hypothesized that this problem could be solved by appropriately increasing the concentration of Calco Oil Red N-1700 added. In this study, we found that the adult flight performance in the 0.5% treatment was significantly reduced, which was unsuitable for field experiments. Similarly, a 0.1% concentration of Calco Oil Red N-1700 significantly reduced the pupation and egg-hatching rates of *H. assulta* [37].

There are many advantages to using Calco Oil Red N-1700 for marking insects. Compared to some physiological and biochemical marking methods (e.g., elemental marking, stable isotope marking, and protein marking), the cost of using this method is low [20,39]. It is also easy to use and is suitable for marking large numbers of insects. The powder is added directly to the larval artificial diet so the insects can feed on it for self-marking, and it is easy to identify after recapture. In addition, this marker is stable. As an internal marker, it is not affected by adverse environmental conditions (heavy rain, high temperature, etc.) and does not transfer to other insects [38]. The in vitro dye marking method also has the advantages of being inexpensive, simple, and easy to identify; however, it may not be as stable as the in vivo dye marking method and is susceptible to loss due to environmental conditions (e.g., heavy rain and dew) in the field or insect movement (e.g., long-distance migration).

Marker persistence is also an important evaluation metric. This study found that the marking colors of females and males persisted for at least five and nine days, respectively. FAW migration begins at a young age (most likely at two days old) and can last for two to three consecutive nights [30]. Therefore, a marking duration of at least five days can basically meet the requirements for MRR experiments. As a result, marked females at older ages may require more sensitive detection methods (e.g., chromatography) for identification [38]; however, this will increase the difficulty and cost of MRR experiments, which typically require a large number of marked individuals. The reason why females have a shorter marking duration than males may be that the coloring location of the markers is mainly the fat bodies, and the mating and oviposition behaviors of females need to consume a lot of energy, resulting in the rapid degradation of fat bodies for females, thus potentially causing the marking color to become lighter or disappear. This study was conducted under laboratory conditions, and further studies are needed to evaluate its performance under field conditions.

In addition to traditional marking methods, many new marking methods, such as genetic markers and transgenic markers, have emerged with the rapid development of molecular biology, transgenic engineering technology, etc. For example, transgenic engineering technology can be used to modify insects to obtain one or more visible genetic markers. Simmons et al. [40] developed a transgenic cotton bollworm strain with fluorescent genetic markers that performed well under field conditions. Green fluorescent protein (GFP) and red fluorescent protein (DsRed) are widely used as biomarker genes [40,41]. These fluorescent proteins naturally emit fluorescence at a specific wavelength and can be continuously passed on to offspring. This marking technique can be applied to various insects. Although genetically modified markers have great potential for application in MRR experiments, successfully developing insects with genetically modified markers is relatively difficult, requires extensive research and high costs, and may also face problems such as complex detection steps and high detection costs [20]. Each marking method has advantages and disadvantages, which must be comprehensively considered according to the actual application context.

This study identified an economically effective method of marking FAWs, which will contribute to the implementation of an MRR project for the pest. Through this project, we can further understand the regional migration behavior of the FAW and estimate the population density and occurrence trend in the field. To reduce the use of chemical pesticides, it is necessary to adopt new green management strategies in newly invaded areas, such as planting resistant varieties and implementing SIT projects [10,42]. It is also valuable for monitoring the evolution of FAW resistance to transgenic crops and formulating appropriate resistance management strategies, and useful for evaluating the feasibility of using SIT to control the pest in the year-round breeding region. Taken together, this study can help formulate effective management strategies for the FAW. However, there are still many difficulties to be overcome in conducting MRR experiments, such as the large-scale rearing and transportation of marked individuals, especially recapture, which is the most difficult and important aspect of MRR experiments.

## 5. Conclusions

*Spodoptera frugiperda* (FAW) has already become a major agricultural pest in many countries of the world. Mark–release–recapture (MRR) is an effective method to study the migration patterns of insects. In this study, we screened a suitable internal marking method for adult FAWs (i.e., adding a 0.2% concentration of Calco Oil Red N-1700 to an artificial diet for larvae to feed on); the marking effect was excellent and did not negatively affect the biological indicators of the FAWs. This work lays the foundation for conducting MRR experiments, and is helpful for forecasting and regional management efforts associated with FAWs.

## Figures and Tables

**Figure 1 insects-15-00561-f001:**
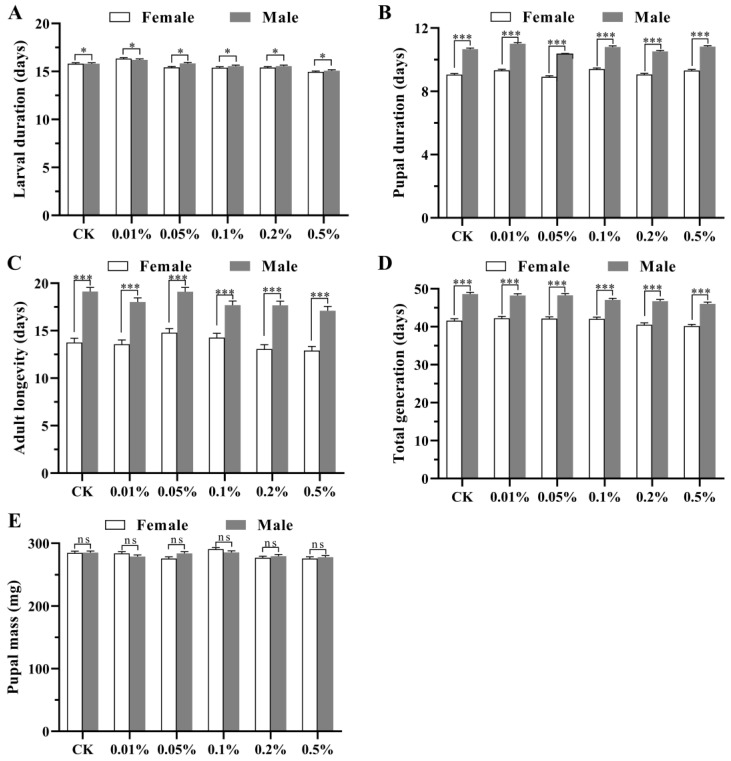
Mean duration (±SE) of developmental stages ((**A**) larval duration; (**B**) pupal duration; (**C**) adult longevity; (**D**) total generation; (**E**) pupal mass) of female and male *Spodoptera frugiperda* individuals fed different diets. CK: normal diet, without Calco Oil Red N-1700. * *p* < 0.05, *** *p* < 0.001, ns *p* > 0.05.

**Figure 2 insects-15-00561-f002:**
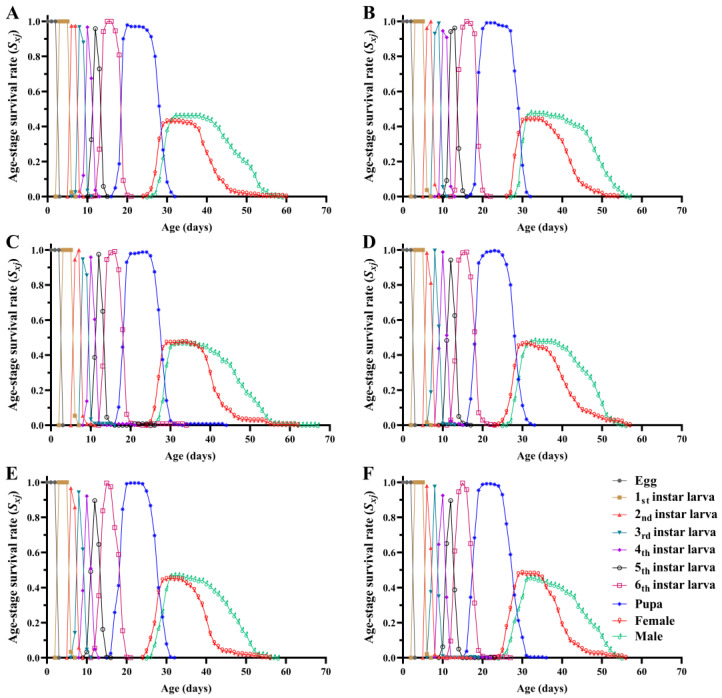
Age–stage survival rate (*s_xj_*) of *Spodoptera frugiperda* fed diets containing different concentrations of Calco Oil Red N-1700. (**A**) Normal diet treatment; (**B**) 0.01% treatment; (**C**) 0.05% treatment; (**D**) 0.1% treatment; (**E**) 0.2% treatment; (**F**) 0.5% treatment.

**Figure 3 insects-15-00561-f003:**
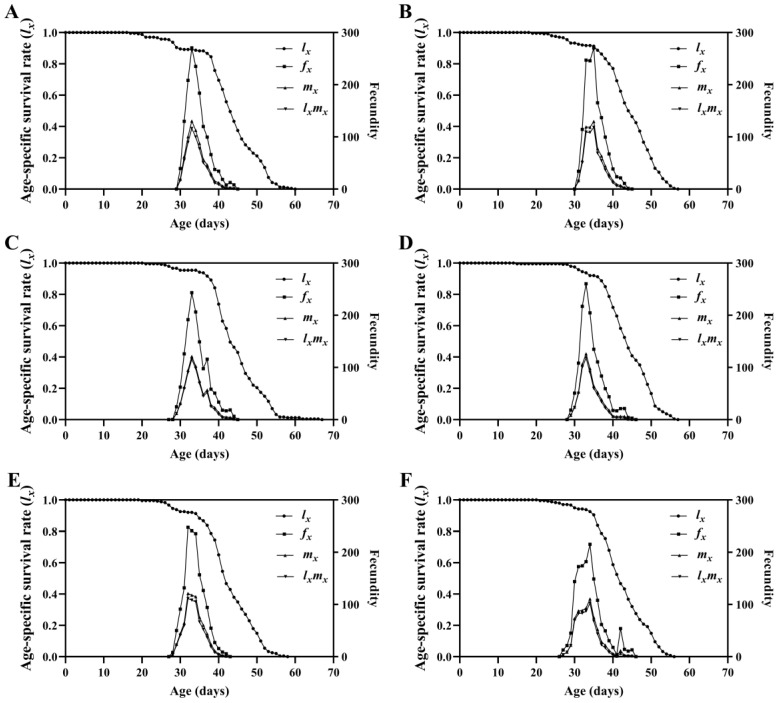
Age-specific survival rate (*l_x_*), age-specific fecundity of female adults (*f_x_*), age-specific fecundity (*m_x_*), and age-specific maternity (*l_x_m_x_*) of *Spodoptera frugiperda* fed diets containing different concentrations of Calco Oil Red N-1700. (**A**) Normal diet treatment; (**B**) 0.01% treatment; (**C**) 0.05% treatment; (**D**) 0.1% treatment; (**E**) 0.2% treatment; (**F**) 0.5% treatment.

**Figure 4 insects-15-00561-f004:**
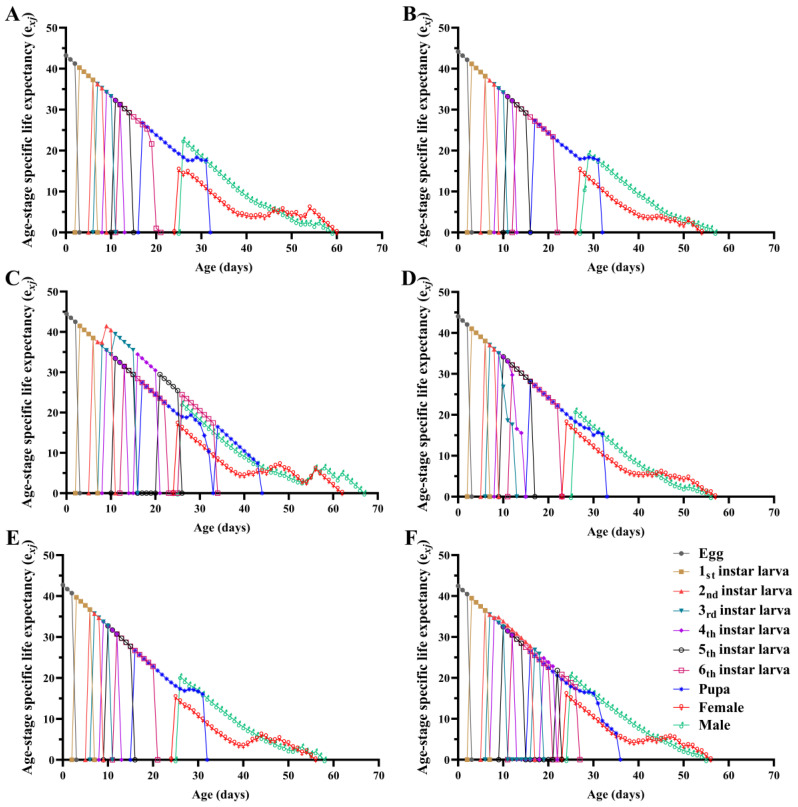
Age–stage specific life expectancy (*e_xj_*) of *Spodoptera frugiperda* fed diets containing different concentrations of Calco Oil Red N-1700. (**A**) Normal diet treatment; (**B**) 0.01% treatment; (**C**) 0.05% treatment; (**D**) 0.1% treatment; (**E**) 0.2% treatment; (**F**) 0.5% treatment.

**Figure 5 insects-15-00561-f005:**
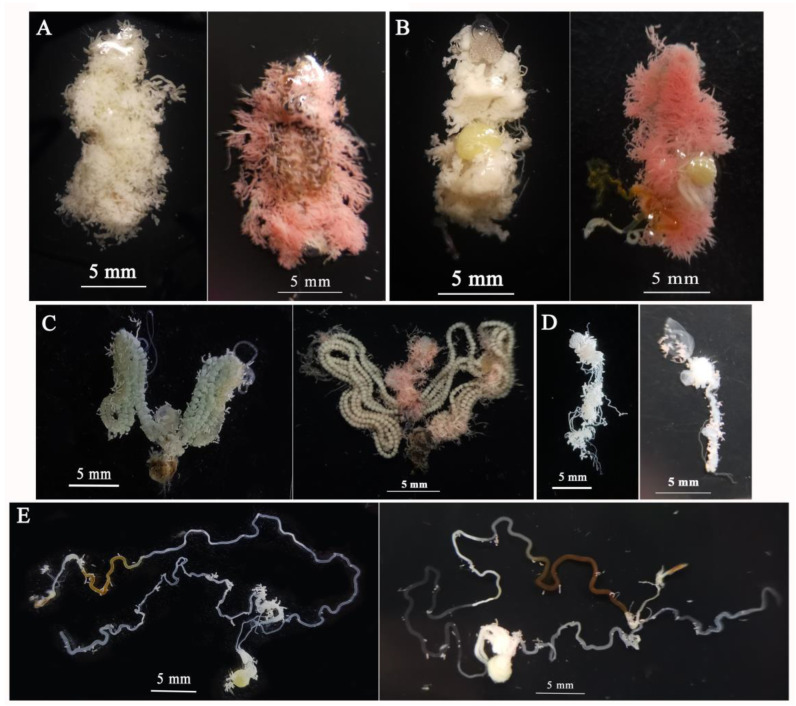
The reproductive system, intestines, and fat bodies of the marked adults of *Spodoptera frugiperda*. (**A**) Abdomen of 1-day-old females; (**B**) abdomen of 1-day-old males; (**C**) reproductive system and intestine of 5-day-old females; (**D**) reproductive system of 1-day-old males; (**E**) intestine of 1-day-old males. All 5 panels show the control adult on the left and the marked adult on the right.

**Figure 6 insects-15-00561-f006:**
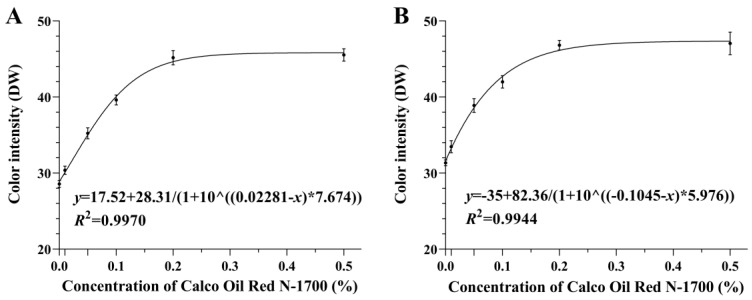
Color intensity (mean ± SE) of the fat bodies of 1-day-old female (**A**) and male (**B**) FAWs fed diets containing different concentrations of Calco Oil Red N-1700.

**Table 1 insects-15-00561-t001:** Developmental period and reproductive performance of *Spodoptera frugiperda* fed diets containing different concentrations of Calco Oil Red N-1700.

Parameters	CK	0.01%	0.05%	0.1%	0.2%	0.5%
Egg (days)	3.00 ± 0.00	3.00 ± 0.00	3.00 ± 0.00	3.00 ± 0.00	3.00 ± 0.00	3.00 ± 0.00
1st instar larva (days)	3.03 ± 0.01 a	3.04 ± 0.01 a	3.05 ± 0.01 a	3.02 ± 0.01 a	3.03 ± 0.01 a	3.02 ± 0.01 a
2nd instar larva (days)	1.98 ± 0.01 ab	2.03 ± 0.01 a	2.02 ± 0.02 a	1.80 ± 0.03 c	1.88 ± 0.03 bc	1.66 ± 0.05 d
3rd instar larva (days)	1.90 ± 0.02 a	1.97 ± 0.01 a	1.88 ± 0.03 a	1.76 ± 0.03 b	1.75 ± 0.03 b	1.72 ± 0.03 b
4th instar larva (days)	1.80 ± 0.03 b	1.93 ± 0.02 a	1.76 ± 0.04 b	1.97 ± 0.03 a	1.86 ± 0.03 ab	1.94 ± 0.03 a
5th instar larva (days)	2.07 ± 0.03 c	2.30 ± 0.03 a	2.11 ± 0.04 bc	2.13 ± 0.03 bc	2.23 ± 0.03 ab	2.05 ± 0.04 c
6th instar larva (days)	5.06 ± 0.03 a	4.97 ± 0.02 ab	4.85 ± 0.04 bc	4.82 ± 0.04 cd	4.70 ± 0.03 de	4.63 ± 0.04 e
Larval duration (days)	15.84 ± 0.04 b	16.25 ± 0.04 a	15.66 ± 0.11 bc	15.49 ± 0.06 c	15.46 ± 0.07 c	15.02 ± 0.07 d
Pupal stage (days)	9.88 ± 0.08 bcd	10.21 ± 0.07 a	9.62 ± 0.06 d	10.12 ± 0.07 ab	9.81 ± 0.06 cd	10.05 ± 0.08 abc
Egg-pupa stage (days)	28.69 ± 0.09 b	29.48 ± 0.08 a	28.26 ± 0.13 cd	28.59 ± 0.10 bc	28.29 ± 0.10 bcd	28.06 ± 0.12 d
Adult longevity (days)	16.51 ± 0.34 ab	15.91 ± 0.35 abc	16.94 ± 0.34 a	16.01 ± 0.34 abc	15.41 ± 0.35 bc	14.94 ± 0.35 c
Total generation (days)	45.20 ± 0.39 ab	45.39 ± 0.37 a	45.20 ± 0.39 ab	44.60 ± 0.37 ab	43.70 ± 0.38 bc	43.00 ± 0.39 c
Pupa mass (mg)	284.89 ± 1.98 ab	281.28 ± 1.60 abc	279.81 ± 1.83 bc	287.90 ± 1.83 a	278.18 ± 2.01 bc	276.77 ± 1.87 c
Pre-oviposition period (days)	4.52 ± 0.15 a	4.78 ± 0.13 a	4.98 ± 0.18 a	4.62 ± 0.16 a	4.39 ± 0.14 a	4.59 ± 0.17 a
Oviposition period (days)	7.34 ± 0.19 a	6.77 ± 0.19 abc	7.08 ± 0.20 ab	6.34 ± 0.17 cd	6.51 ± 0.19 bcd	5.86 ± 0.13 d
Eggs deposited per female (*n*)	1596.32 ± 42.59 a	1523.92 ± 40.09 a	1509.56 ± 40.62 a	1544.89 ± 51.34 a	1503.51 ± 44.34 a	1426.86 ± 44.43 a
Mating rate (%)	89.32 ± 4.08 a	90.96 ± 3.46 a	89.97 ± 2.34 a	85.43 ± 1.25 a	94.71 ± 4.01 a	97.06 ± 1.70 a

Data are means ± SE, values in the same row, followed by different lowercase letters indicating a significant difference among different diets (one-way ANOVA, Tukey’s HSD; *p* < 0.05). CK: normal diet, without Calco Oil Red N-1700.

**Table 2 insects-15-00561-t002:** Larval and pupal survival and adult deformity rates of *Spodoptera frugiperda* fed diets containing different concentrations of Calco Oil Red N-1700.

Parameters	CK	0.01%	0.05%	0.1%	0.2%	0.5%
Larval survival rate (%)	97.92 ± 0.83 b	99.58 ± 0.42 ab	100 ± 0.00 a	99.58 ± 0.42 ab	100 ± 0.00 a	100 ± 0.00 a
Pupal survival rate (%)	91.47 ± 0.93 a	94.14 ± 1.13 a	96.25 ± 0.72 a	97.07 ± 0.43 a	94.17 ± 2.20 a	97.08 ± 1.82 a
Adult deformity rate (%)	0.94 ± 0.47 a	2.25 ± 1.18 a	0.43 ± 0.43 a	0.87 ± 0.43 a	1.83 ± 1.23 a	2.14 ± 0.85 a

Data are means ± SE, values in the same row followed by different lowercase letters indicate a significant difference (one-way ANOVA, Tukey’s HSD; *p* < 0.05). CK: normal diet, without Calco Oil Red N-1700.

**Table 3 insects-15-00561-t003:** Life table parameters (mean ± SE) of *Spodoptera frugiperda* fed diets containing different concentrations of Calco Oil Red N-1700.

Parameters	CK	0.01%	0.05%	0.1%	0.2%
Net reproductive rate *R*_0_ (offspring/individual)	598.62 ± 52.02 a	609.57 ± 51.20 a	635.28 ± 51.11 a	585.77 ± 52.08 a	638.99 ± 51.42 a
Mean generation time *T* (d)	34.59 ± 0.15 b	35.56 ± 0.15 a	34.42 ± 0.18 bc	34.36 ± 0.18 bc	33.92 ± 0.18 cd
Intrinsic rate of natural increase *r* (d^−1^)	0.1848 ± 0.0027 a	0.1803 ± 0.0026 b	0.1875 ± 0.0026 ab	0.1855 ± 0.0028 ab	0.1905 ± 0.0027 a
Finite rate of increase *λ* (d^−1^)	1.2030 ± 0.0033 ab	1.1976 ± 0.0031 b	1.2063 ± 0.0032 ab	1.2038 ± 0.0034 ab	1.2098 ± 0.0033 a

Data in same row followed by different letters significantly differed (paired bootstrap test, *p* < 0.05). CK: normal diet, without Calco Oil Red N-1700.

**Table 4 insects-15-00561-t004:** Flight performance (mean ± SE) of *Spodoptera frugiperda* fed diets containing different concentrations of Calco Oil Red N-1700, recorded in a laboratory using tethered-flight mills. Flight performance is expressed using multiple parameters (i.e., distance, duration, and mean velocity).

Parameters	CK	0.01%	0.05%	0.1%	0.2%	0.5%
Flight distance (km)	20.61 ± 1.53 ab	16.88 ± 1.91 ab	20.59 ± 1.99 ab	21.62 ± 2.92 a	24.91 ± 2.17 a	12.28 ± 2.05 b
Flight duration (h)	6.49 ± 0.44 a	5.45 ± 0.51 ab	6.65 ± 0.54 a	6.07 ± 0.60 a	7.16 ± 0.48 a	3.35 ± 0.51 b
Flight velocity (km/h)	3.19 ± 0.13 a	2.95 ± 0.15 a	3.13 ± 0.17 a	3.46 ± 0.22 a	3.40 ± 0.16 a	3.53 ± 0.24 a

Different letters in same column indicate significantly different flight parameters (one-way ANOVA, *p* < 0.05, Tukey’s HSD). CK: normal diet, without Calco Oil Red N-1700.

**Table 5 insects-15-00561-t005:** The coloring rate of adults fed diets containing 0.2% of Calco Oil Red N-1700 at different ages.

	1-Day-Old	3-Day-Old	5-Day-Old	7-Day-Old	9-Day-Old
Female (%)	100 ± 0.00 a	100 ± 0.00 a	100 ± 0.00 a	85.19 ± 2.67 b	65.56 ± 2.94 c
Male (%)	100 ± 0.00 a	100 ± 0.00 a	100 ± 0.00 a	100 ± 0.00 a	100 ± 0.00 a

Data are means ± SE. Different letters in same column indicate significantly different flight parameters (one-way ANOVA, *p* < 0.05, Tukey’s HSD).

## Data Availability

The original contributions presented in the study are included in the article; further inquiries can be directed to the corresponding author.

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
