# Peer review of "An Internal Marking Method for Adult Spodoptera frugiperda Smith Using an Artificial Diet Containing Calco Oil Red N-1700"

_insects, 2024, doi:10.3390/insects15080561_

Round 1
Reviewer 1 Report
Comments and Suggestions for Authors
REVIEWER Insects 310788
This manuscript traits a priority quarantine pest as Spodoptera frugiperda, that damage a lot of crops in Europe, Africa and Asia. The high dispersal capacity of S. frugiperda, even 100 km in one night, with the high reproductive ability, several annual generations, determine its very high capacity of destructiveness of the crops.
As reported in the EFSA survey card, in Europe many efforts are being made to monitor the emergency situation on this priority pest, standardising the methods of surveillance, with annual surveys at the most risky survey sites, and so on. In addition, it is suggested to join the online tool developed by the United Nations FAW Monitoring and Early Warning System (FAMEWS) in order to monitor the spread of the moth.
Certainly, the results obtained under laboratory conditions may provide other data for dealing with this phytosanitary emergency. The authors start from the hypothesis that Calco Oil Red N-1700 is a marker that can produce highly visible and durable colors when added to the diets of some lepidopteran insects by using in Mark–release–recapture (MRR) method. By using a MRR in lab conditions the authors can be studied the growth, development, survival, reproduction, and flight performance, in order to develop an experimental design in field condition to study the migration pattern of the FAW.
The manuscript is well done and structured; the experimental design and formal analysis are precise. The discussion take in consideration all the aspects of the experiment, with a good discussion with the recent literature. My suggestion is to add the detailed data obtained in Europe by using a annual surveillance activity of the Member States, and collect in EFSA pest survey card https://storymaps.arcgis.com/stories/06fb4d48431a409eadfa2413544d275e
As the authors reported in the conclusion, the MRR protocol is very difficult to apply in the field, because the few frequent or rare events of recapture and low capacity by the operators to follow the butterflies.
I believe that the manuscript is to be accepted with minor revision.
The authors can find my suggestions and revision directly in the pdf as comments.
LINE 42 Change flight performance or flight activity
LINE 193 ADD the meaning of the lx, fx, mx, lxmx, and for the demographic parameters
LINE 236 Table 1 Suggest to review the format of the table 1; there are a lot of information but itis very difficult to check it. Maybe create a rows with different colour grey, white or dark grey, or separate the different part of the results by lines of by different colour or etc: one for developmental and one for reproductive performance
Lines 241 correct the paranthesis open after generation
Lines 290 Table 3 see the comment for table 1
Lines 533 Correct the dot with comma after the year of publication.
Reviewer 2 Report
Comments and Suggestions for Authors
Comments for authors
Line 15: Spodoptera frugiperda Smith (FAW). The abbreviation FAW must be added after its representative full name i.e. Fall armyworm.
Line 49-50: It is estimated that the FAW can cause an average pest-related yield loss of 12%. Where these losses were reported? I mean globally or on a specific region, also mention crop? Please be specific with this statement.
Line 56-57: FAW populations from multiple countries or regions can communicate with each other. How they communicate? Please rewrite this sentence for more clarity.
Line 61: Zhou et al. Better to mention reference number after et al. [16]. Same applies with Wu et al. in line 63. Showers et al. [19] line 70. Line 357: Li [37]. Line 363: Hagler et al. [38].
Line 65. Please write as (increases the complexity associated with its forecasting and regional management). Also add a citation here.
Line 147: Three replicates can be questionable but given 80 individuals in each replicate, it is fine.
Line 183: This was followed by Tukey’s HSD post-hoc test. What is the purpose of using this test? Please mention it is used for separating treatment with significant differences.
Table 1: values after ± indicates SE or SD. Please mention them in table footnotes. Same implies with rest of tables and figures. Why this value is so high in Eggs deposited per female? Please recheck your analysis. Also, what is CK in table 1. Please elaborate in table and figures foot notes.
Figure 1: Y-axis, please write days instead of day. Why lettering is missing in Figure 1?
Figure 2: X-axis, please write days instead of day. Same implies with rest of figures.
Line 352: for some insect species. Mention their names. Whether these are also lepidopterous insects?
Line 431: Please write: regional management efforts associated with FAW.
